# Expanding Canonical Spider Silk Properties through a DNA Combinatorial Approach

**DOI:** 10.3390/ma13163596

**Published:** 2020-08-14

**Authors:** Zaroug Jaleel, Shun Zhou, Zaira Martín-Moldes, Lauren M. Baugh, Jonathan Yeh, Nina Dinjaski, Laura T. Brown, Jessica E. Garb, David L. Kaplan

**Affiliations:** 1Department of Biomedical Engineering, Tufts University, 4 Colby St, Medford, MA 02155, USA; zarougj@bu.edu (Z.J.); shun.zhou@genscript.com (S.Z.); Zaira.Martin_Moldes@tufts.edu (Z.M.-M.); lbaugh@mit.edu (L.M.B.); yeh.jona@gmail.com (J.Y.); ndinjaski@partners.org (N.D.); Laura.Brown@milliporesigma.com (L.T.B.); 2School of Medicine, Boston University, Boston, MA 02118, USA; 3National Engineering Laboratory for Modern Silk, College of Textile and Clothing Engineering, Soochow University, Suzhou 215123, China; 4Department of Biological Engineering, Massachusetts Institute of Technology, 21 Ames St #56-651, Cambridge, MA 02142, USA; 5Department of Process Development, Akouos Inc., 645 Summer St. Boston, MA 02210, USA; 6Division of Innovation, Partners HealthCare Innovation, 215 First Street, Cambridge, MA 02142, USA; 7Department of Research Solutions North America, MilliporeSigma, 400 Summit Dr, Burlington, MA 01803, USA; 8Department of Biological Science, University of Massachusetts Lowell, 198 Riverside Street, Olsen Hall 234, Lowell, MA 01854, USA; Jessica_Garb@uml.edu

**Keywords:** biomaterials, recombinant spider silk, broadening silk properties

## Abstract

The properties of native spider silk vary within and across species due to the presence of different genes containing conserved repetitive core domains encoding a variety of silk proteins. Previous studies seeking to understand the function and material properties of these domains focused primarily on the analysis of dragline silk proteins, MaSp1 and MaSp2. Our work seeks to broaden the mechanical properties of silk-based biomaterials by establishing two libraries containing genes from the repetitive core region of the native *Latrodectus hesperus* silk genome (Library A: genes *masp1*, *masp2*, *tusp1*, *acsp1*; Library B: genes *acsp1*, *pysp1*, *misp1*, *flag*). The expressed and purified proteins were analyzed through Fourier Transform Infrared Spectrometry (FTIR). Some of these new proteins revealed a higher portion of β-sheet content in recombinant proteins produced from gene constructs containing a combination of *masp1*/*masp2* and *acsp1*/*tusp1* genes than recombinant proteins which consisted solely of dragline silk genes (Library A). A higher portion of β-turn and random coil content was identified in recombinant proteins from *pysp1* and *flag* genes (Library B). Mechanical characterization of selected proteins purified from Library A and Library B formed into films was assessed by Atomic Force Microscopy (AFM) and suggested Library A recombinant proteins had higher elastic moduli when compared to Library B recombinant proteins. Both libraries had higher elastic moduli when compared to native spider silk proteins. The preliminary approach demonstrated here suggests that repetitive core regions of the aforementioned genes can be used as building blocks for new silk-based biomaterials with varying mechanical properties.

## 1. Introduction

Materials inspired by nature are of interest for insights into structure–function relationships, with spider silks as a canonical example [1,2,3,4,5]. One focus in biomaterials research is to artificially replicate spider silk proteins [1,3,6,7,8,9]; in contrast to the silkworm, *B. mori*, the spider’s aggressive territorial behavior and its cannibalism render spider farming not feasible [10,11]. Owing to their distinct mechanical properties, including outstanding elasticity, high tensile strength and superior toughness, spider silks could be useful in medical and industrial fields [10,12,13,14,15,16,17]. Recent advances in genetic engineering have led to increased insight into silk proteins and the structural organization of spider-silk-encoding genes [17,18]. Hence, more research focused on the bioengineering technology to develop artificial spider silks comparable to natural spider silks mostly use bacteria [11,15], while yeast [19,20], mammalian cells [21], plants [22] and insect cells are also used as platforms. However, the cloning of spider silk genes presents both challenges and opportunities due to the highly repetitive nature of the native proteins known as spidroins [8,23].

The evolutionary success of spiders over the past 400 million years is due in part to the great versatility of silks [15,24]. Spiders require silk for a multitude of functions in their daily lives from casting their webs [25], to catching prey [26], covering eggs [27] and using draglines [28]. Certain amino acid motifs within the repetitive core region of different spidroins that compose the distinct silk types have been identified as responsible for some of the specific properties of silk (Figure 1) [2,29].

Orb-weavers and cob-web weaving spiders such as *Latrodectus*, can produce up to seven functionally distinct silk types, each of which is primarily composed of silk-specific spidroins. Dragline silk, produced by major ampullate glands, is primarily made of Major ampullate spidroin 1 and 2 (genes *masp1* and *masp2*, respectively), which play a role in the formation of tightly knit β-sheet structures, likely due to the presence of peptide motifs of poly-alanine domains (poly (A/GA)), and glycine rich domains (GGX) with X being either Y, L or Q, facilitating 3_10_ helix formation [30]. These repetitive core motifs also play a role in the high tensile strength and high Young’s modulus of the dragline silk [28,31]. In addition, a third MaSp3 type isolated from *Argiope argentata* and *L. hesperus* [18], and a fourth MaSp4 type isolated from *Careostris darwini* [32] were identified. MaSp3 ampullate spidroins lack polyalanine and glycine-proline-glycine motifs typically present in MaSp1 and MaSp2, and the repetitive regions contain larger and more polar amino acids than that in MaSp1 or MaSp2 [33]. The MaSp4 spidroin lacks polyalanine motifs and includes a novel GPGPQ motif; proline abundance and arrangement in GPGPQ motifs may increase dragline extensibility by forming novel structural domains embedded among other MaSp proteins or by packing in more β-turns per protein monomer [32]. The minor ampullate spidroin 1 (*misp1*) gene encodes proteins in prey-wrapping and auxiliary spiral silk [34]. MiSp1 shares many properties with the MaSp spidroins including the presence of GGX and poly(A) motifs, and minor ampullate silk mechanically behaves similarly to dragline silk—both a high Young’s modulus and strength-but has a superior extensibility [35]. The aciniform spidroin 1 (*acsp1*) gene is responsible for the egg case and prey wrapping silk that must be hard and durable to withstand environmental stress [36,37]. Mechanically, AcSp1-based aciniform silk contains a higher Young’s modulus and extensibility compared to Masp-containing dragline silk but lacks in tensile strength [17]. Tubuliform spidroin 1 (*tusp1*) constitutes the hard outer casing of spider eggs [37]. TuSp1 is unique in that it contains β-sheet structures through the presence of glutamine and serine interspersed in poly A regions [38], which facilitate greater spacing of the resulting β-sheets compared to MaSp1 and MaSp2 poly A/GA regions. Pyriform spidroin 1 protein (*pysp1*) comprises the adhesive attachment disks that help anchor the dragline and the webs to substrates [25,39]. Although not much is known about the mechanical properties of this class of spidroin proteins, primary sequence motifs in PySp1 have identified large stretches of alternating proline (Px) or glutamine (QQxxxx) motifs that are suggested to play a role in adhesion and elasticity [40]. Finally, Flag silk (*flag*) comprises the capture spiral of a spider web and thus must be flexible and elastic [41,42]. The characteristic extensibility of Flag is that it can stretch up to 200% of the original length with a low Young’s modulus due to the presence of several GPGXX motifs that generate a β-spiral motif comprised of two linked type II β-turns [17,43]. 

In *Nephila* and other orb-weavers, the relationship to silk types and their function is better known. *Latrodectus* has the same spidroins, but in some cases, the functional use is the same, and for other silks (such as *flag*), it is not fully known. The gene designs used in this study, except for *flag*, were derived from sequenced cDNAs from the black widow spider *L. hesperus*. The repetitive core genes from *L. hesperus* were used due to the superior mechanical properties of native *L. hesperus* silk in both toughness and strength compared to other spider species [28]. However, due to the superior elasticity of *N. clavipes* Flag silk [41], the cDNA from the repetitive core region of this specie was used as the basis for the *flag* gene design used in our library study. 

The main focus for structure–function analyses of spider silks to date has been on the use of individual spidroin proteins, though a few studies have examined chimeras composed of two spidroin types as the core repeats. By manipulating both the ratio of different motifs and combining different repetitive motifs derived from various spidroins, it is possible to design novel synthetic model proteins to understand and probe a broader range of artificial silk properties, including new specific biological materials with tunable mechanical properties. Examples of chimeric spider silk proteins include MaSp1/MaSp2 fusion proteins [44,45,46], MaSp1/Flag [47], MaSp2/Flag [48,49] or MaSp/AcSp [50]. Another example is the generation of chimeric silkworm–spider silks through expression in transgenic silkworms. This strategy was successful in the expression of dragline silk [51,52,53] and minor ampullate genes [53] in silkworms.

The current project focused on the relatively understudied repetitive core regions of the egg case silk genes (*tusp1*, *acsp1*), the structural web genes (*pysp1*, *misp1*, *flag*) and the two long-known dragline genes (*masp1*, *masp2*) as the building blocks for the synthesis of new protein-based biomaterials through the use of DNA libraries. Library A was designed to generate proteins with high toughness and tensile strength due to the combined material properties of the egg case silk components and the dragline silk components, respectively. In contrast, Library B was designed to generate new spider silklike proteins with more elastomeric features, due to the elastic and adhesive properties of the respective genes and gene products. Through this strategy of biased library building blocks, we sought to broaden the range of mechanical properties of silk-based biomaterials. Mechanical and secondary structural properties of selected protein chimeras were analyzed, resulting in the identification of three recombinant proteins with varied and unique Young’s moduli and secondary structural motifs. Secondary structural properties were characterized through Fourier Transform Infrared Spectrometry (FTIR) to analyze β-sheet content, while the surface morphology and mechanical properties were analyzed using Atomic Force Microscopy (AFM) to determine the Young’s Modulus (E) as a measure of a materials stiffness or resistance to deformation.

## 2. Materials and Methods

### 2.1. pET19b4 Construction

All molecular biology procedures were performed following standard protocols [54]. Briefly, the parent plasmid pET19b3 [55] was cleaved using the restriction enzymes NdeI and BamHI. Annealing of the b4 sense and antisense linker (5′ TATGGCTAGCGGGCTCACTAGTTAAG 3′) created the b4 insert that contained one restriction enzyme site for BanII as well as restriction enzyme sites for NheI and SpeI to allow for head-to-tail ligation of monomeric inserts obtained from the original screens, which was then ligated into the pET19b. All enzymes used in this project were purchased from New England Biolabs (Ipswich, MA, USA).

### 2.2. Plasmid and Library Construction

DNA sequences for each of the seven genes were chemically synthesized from GenScript (Piscataway, NJ, USA) and designed to encode for the repetitive core region (Table 1). The resulting DNA sequences were then isolated from the pUC57 plasmid using BanII restriction enzyme, purified and ligated and subsequently cloned into the pET19b4 expression vector for establishment of the library. For the creation of repeated DNA sequences such as A10_2_ and B10-22-17_2_, head-to-tail ligation of monomeric recombinant inserts was achieved through the NheI and SpeI restriction sites. All the resulting plasmids containing the combined spidroin genes were then transformed into competent DH5α *Escherichia coli* cells (New England Biolabs, Ipswich, UK). Robust screening of 652 colonies for inserts greater than 1 kb revealed 15 total colonies with unique insert arrangements. 

### 2.3. Expression and Purification of Recombinant Proteins

Plasmids obtained from the recombinant DNA library were transformed into competent *E. coli* BL21* (New England Biolabs, Ipswich, MA) for expression as described in [56]. Briefly, transformants were cultured in a 1 L hyper-broth media (Sigma-Aldrich, Natick, MA, USA) containing a yeast extract medium 25 g/L and a 15% glucose nutrient mix as well as ampicillin (100 µg/mL) at 37 °C under aeration conditions until cultures reached the mid logarithmic growth phase (OD600 = 0.5–0.8); protein expression was induced by the addition of 1 mM Isopropyl-β-d-thiogalactopyranoside (IPTG) (Sigma-Aldrich). After 6 h, the culture was centrifuged at 7500× *g* for 20 min, and the resulting pellet was collected and stored at −20 °C. The pellet was lysed in 8 M urea pH = 8.0 overnight after which Nickel-NTA beads (Qiagen, Hilden, Germany) were added. Protein column purification was performed using urea buffered at decreasing pH intervals = 8.0, 6.3, 5.9, 4.5 with the targeted protein being eluted at pH 4.5. The resulting elutants were run on an SDS-PAGE. The aliquots containing the target proteins were prepared for dialysis in 50 mM phosphate buffer at pH 5.4, and finally, water to remove the urea and induce proper folding of the proteins. The resulting solution was then lyophilized to obtain the pure protein in solid form. 

### 2.4. Protein Films and Characterization

Purified lyophilized protein samples were dissolved in hexafluoro-2-propanol (HFIP) (2.5% w/v) (Sigma-Aldrich) at 37 °C overnight to dissolve the silk proteins. Six films of each sample were prepared by depositing 30 µL of each dissolved protein solution onto different polydimethylsiloxane (PDMS) disks (R = 6 mm). The solvent was removed by evaporation in a fume hood at room temperature (RT) for 1 h. To induce β-sheet formation, the films were immersed in 70% methanol overnight at RT and then air dried.

### 2.5. Fourier Transform Infrared Spectroscopy

Fourier Transform Infrared Spectroscopy (FTIR) of the films was carried out with an Attenuated Total Reflectance FTIR machine (Jasco, Oklahoma City, OK, USA). Deconvolution of the peaks was performed with PeakFit software provided by SigmaPlot. Ten Gaussian peaks were selected across the amide I absorption band (1720–1580 cm^−1^). Four secondary structure motifs were analyzed corresponding to β-sheet (1618–1629 cm^−1^ and 1697–1703 cm^−1^), α-helix (1658–1667 cm^−1^), β-turns (1668–1696 cm^−1^) and random coil (1630–1657 cm^−1^) [56]. Using the second derivative method, the relative area of the curves corresponding to each of the secondary structural motifs was calculated; relative areas were divided by the summed area of all the deconvoluted curves and transformed into percentages.

### 2.6. Analysis of Primary Sequence Motifs

The presence of GPGXX, GGX, poly A/GA and poly AAQAA/AASQSA motifs in the translated sequence data of selected constructs (B10-22-17_2_, A10_2_, A261, A219, B10-22-17) were identified using the alignment tool in Basic Local Alignment Search Tool (BLAST) from the NCBI [57].

### 2.7. Atomic Force Microscopy (AFM) for Mechanical Properties

Lyophilized proteins A10_2_ (Masp1-Masp2-Masp1-Acsp1)_2_, A261 (Tusp1_2_-Acsp1-(Masp1)_2_) and B10-22-17_2_ ((Pysp1)-_3_-Flag-Pysp1-MiSp1-Pysp1-(Pysp1_2_-Acsp1)_2_) were dissolved overnight in HFIP (2.5% w/v) and cast on precleared glass slides at RT and allowed to dry in a vacuum chamber to minimize bubble formation. A Veeco (Town of Oyster Bay, New York, NY, USA) Dimension 3100 Atomic Force Microscope was used to take force-displacement measurements over a 20 × 20 μm area using a 16 × 16 grid. A 5 μm diameter borosilicate glass beaded probe with a 0.06 N/m spring constant (Novascan, Ames, IA, USA) was used to record all modulus data. To avoid interference with the underlying glass coverslip, a 750 nm ramp depth was used for all measurements. Average roughness values were calculated using Gwyddion (v2.56); the roughness for these samples was calculated in the range 60–300 nm for the different samples analyzed and are summarized in Appendix A. Using the force-displacement curves, the Young’s Modulus was calculated using the Hertz model [58,59,60,61] assuming a Poisson’s ratio of 0.5 because protein networks such as silk obey rubber elasticity. The calculations to determine Young’s modulus were performed using a MATLAB script (R2019a). Topographical images of each of the samples were taken using a non-contact tip (model FESP, Bruker, Billerica, MA, USA). Representative images are 20 × 20 μm and were processed using Nanoscope Analysis 1.5 software. 

### 2.8. Statistical Analysis

Quantitative analysis was performed at least in triplicate and average values were plotted with error bars representing the standard deviation. Comparisons between k > 2 samples were performed through a one-way ANOVA and a post-hoc Tukey Kramer test. Statistical analysis between two treatments of the same sample was performed using a two-sample paired students *t*-test of the means. All statistical analyses had a significance level of *p* < 0.05.

## 3. Results

### 3.1. Construction of Dynamic Repetitive Core Library

Establishment of the two libraries was conducted according to the material properties being pursued. The first library consisted of *masp1*, *masp2*, *tusp1* and *acsp1* core genes from *L. hesperus* and was focused on generating new silk-like protein materials with high stiffness and toughness, while the second library consisted of *misp1*, *acsp1*, *pysp1* and the *flag* core genes and was focused on generating elastic materials with low stiffness and elastomeric features. *acsp1* was included in both libraries because it contains a high Young’s modulus and toughness [35] useful for the stiffness and strength properties desired in Library A recombinants as well as a high extensibility [35] useful for providing the tough yet elastic properties desired in Library B. The core amino acid motifs selected for this work are shown in Table 1. All sequences for the project were selected from *L. hesperus*, except for the *flag* gene where the *N. clavipes* gene was selected.

The seven genes were multimerized upon randomized ligation of the repetitive core genes using a “concatemerization” strategy and cloned into a newly synthesized pET19-b4 expression vector (Figure 2A). Transformants were screened for the presence of library inserts; 652 colonies were screened and 15 were selected for further investigation (including 9 from Library A and 6 from Library B) due to their varied gene patterns and their higher molecular weights (Table 2). 

The subsequent generation of targeted recombinants (A10_2_, B10-22-17_2_) with the aim of forming higher molecular weights similar to native spider silks was carried out using a directional polymerization approach. Three major polymerized recombinants were selected for this approach—A10_2_ (MaSp1-MaSp2-MaSp1-Acsp1)_2_, A261 (TuSp1_2_-AcSp1-MaSp1_2_) and B10-22-17_2_ ((PySp1)_3_-Flag-PySp1-MiSp1-PySp1-(PySp1_2_-AcSp1)_2_). In total, fourteen constructs (A10, A11, A15, A46, A52, A217, A219, A261, A10_2_, B10, B12, B17, B10-22-17, B10-22-17_2_) were successfully produced, purified and their approximate molecular weights determined by SDS-Page confirming a varied combination of inserts of different molecular weights between 22 and 88 kDa (Figure 2B). The yield of the expressed proteins ranged between 50 and 100 mg/L.

### 3.2. Secondary Structure of Library A and Library B generated Proteins

In order to analyze the secondary structure, FTIR analysis was performed before and after methanol addition in the selected constructs: A10, A11, A15, A46, A52, A217, A219, A261, A10_2_, B10, B12, B17, B10-22-17, B10-22-17_2_. FTIR spectra were analyzed across the Amide I absorption band (1580–1720 cm^−1^), and four major secondary structural motifs were analyzed (α-helices, β-sheets, β-turns, random coils). The representative FTIR spectra for untreated and treated A10_2_ films is provided in Appendix A, as well as a representative deconvoluted spectrum for untreated and treated A10_2_ films in Appendix A. In general, the addition of methanol provoked a decrease in the Gaussian peak size for each recombinant protein at 1650 and 1660 cm^−1^ with a corresponding increase in peak size at 1620 cm^−1^ which correlated with a decrease in random coil and α-helix and an increase in β-sheet (Figure 3, Appendix A). 

A219, consisting of exclusively dragline silk, showed a significantly lower β–sheet content (16.8%) and a significantly higher β-turn composition (22.8%) (Figure 3A,C). A10_2_, composed of dragline and aciniform silk, contained the highest amount of GGX motifs as well as one of the highest β-turn percentages amongst Library A (21.2%) (Figure 3C, Table 3). A261, comprised of tubuliform, aciniform and dragline silk, contained the highest percentage β-sheet amongst all analyzed recombinants (34.4%) (Figure 3A). B10-22-17, constituted by pyriform, flagelliform, minor ampullate and aciniform silk, had one of the smallest uninduced β-sheet compositions (8.1%) (Figure 3A), yet the largest methanol-induced fold change in β-sheet percentage of over 200% and the largest percentage of β-turn composition amongst both libraries (26.9%) (Figure 3C). B10-22-17_2_, comprised by pyriform, flagelliform, minor ampullate and aciniform silk, on the other hand, had the reverse trend with one of the highest uninduced and induced β-sheet percentages (29.0% and 32.5%, Figure 3A) and one of the lowest β-turn percentages (11.6%, Figure 3C) amongst Library B recombinants. 

Primary sequence motifs were further analyzed from the translated sequencing data of five selected constructs (B10-22-17_2_, A10_2_, A261, A219, B10-22-17) due to their similar size and varying domains expressed. The presence of GPGXX motifs that correlate with elastic beta spiral structures, GGX motifs that correlate with amorphous 3_10_ helix domains and poly A/GA and AAQAA/AASQSA motifs that correlate with crystalline β-sheet structures and the observed FTIR secondary structures after methanol treatment were analyzed and are summarized in Table 3. For B10-22-17_2_, 8 GPGXX motifs, 4 GGX motifs and 4 poly GAn motifs were identified. For A10_2_, 4 GPGXX motifs, 16 GGX motifs, 6 poly A motifs and 4 poly GAn motifs were identified. For A261, no β-spiral motifs were identified, 9 GGX motifs, 2 poly A motifs, 2 poly GA motifs, 2 AAQAA motifs and 4 AASQSA motifs were identified as the only constructs containing AAQAA/AASQAA motifs. For A219, 6 GPGXX motifs, 11 GGX motifs, 7 poly A motifs and 4 poly GA motifs were identified. Finally, for B10-22-17, 8 GPGXX motifs, 3 GGX motifs and 4 poly GAn motifs were identified.

### 3.3. Mechanical Properties 

To study the surface topography and mechanical properties of the recombinant proteins, images and nano-indentation studies were carried out by AFM. Nano-indentation was performed by collecting force-displacement measurements, and a representative curve of the load-displacement curve is provided as Appendix A. Three representative proteins were chosen—A261, A10_2_ and B10-22-17_2_—based on: (a) high expression efficiencies in BL21* *E. coli*, (b) highest molecular weights obtained to try to approach as much as possible that of native spider silk proteins, (c) high β-sheet content which suggested enhanced material properties and (d) diverse composition of the original monomeric proteins (Table 2). 

Images generated using the AFM showed smooth and uniform surfaces for B10-22-17_2_ films, both before and after methanol treatment. Whereas, for A261 and A10_2_, a smoother surface was shown after methanol treatment, and spherical structures were observed before methanol treatment (Figure 4A). 

Nano-indentation studies showed that the two Library A samples A261 and A10_2_, despite having a lower molecular weight, had a higher elastic modulus than B10-22-17_2_. For B10-22-17_2_, the addition of methanol significantly increased the Young’s Modulus of the resulting films from 3.4 to 4.0 GPa (Figure 4B). However, for A261 and A10_2,_ the presence of methanol decreased the Young’s modulus of the films from 24.1 and 31.7 GPa, to 15.1 and 20.0 GPa, respectively (Figure 4B). 

## 4. Discussion

The establishment of two DNA libraries was conducted with a focus on generating novel biomaterials by combining different repetitive spider silk protein core modules present within the *L. hesperus* and *N. clavipes* genomes. While significant sequence homology of spidroins is conserved across many spider species, *L. hesperus* is a useful model for understanding spidroin repetitive architecture, because unlike most species, several full length or near full-length spidroins have been characterized from its genome [25,28,37,62,63]. These higher order units consist of stretches of different genetic sequences encoding amino acid motifs including GGX and poly A repeats and can be from 70 to 2000 amino acids in length. Likewise, small differences in amino acid sequences, such as high levels of S and Q in tubuliform silk, can contribute to the enhanced material properties of *L. hesperus* silk which are reflected in the sequences used for library design and construction. The library proteins chosen were screened to contain a diverse array of incorporated motifs and a degree of robustness aimed to mimic varied native silk-like properties such as high extensibility and low stiffness, as well as versions with high stiffness and high toughness. 

We performed FTIR to assess the secondary structure of films generated from different silk protein chimeric constructs. Deconvolution of the amide I spectra showed that the addition of methanol resulted in a decreased peak at 1650 and 1660 cm^−1^ with an increased peak at 1620 cm^−1^ suggesting a change in structure from random coils and α-helix to a more ordered β-sheet structure (Figure 3). This remains consistent with expectations, as the addition of methanol has been shown to induce β-sheet formation in many silk-related polypeptides [30,56]. The constructions from the library showed similar secondary structure compositions when compared to native spidroin proteins, as reported in previous publications. For example, native major ampullate silk, from which MaSp1 and MaSp2 were derived, is comprised of approximately 34% β-sheet structure content [6]. In Library A constructs, most of the recombinant proteins contained β-sheet concentrations near this value with a mean value of 31.3% (Figure 3A). Conjugation of dragline silk recombinant construct motifs with egg case components increased the percentage of β-sheet in the resulting protein films as seen by the difference between A219, which consisted of solely dragline silk genes, and the rest of library A, which contained some combinations of dragline and egg case silk. This could be due to the fact that MaSp sequences encoded for one β-sheet motif only at the end of the repetitive domain as well as β-spirals and amorphous 3_10_ helices that would contribute to a higher β-turn and α-helix contribution. In contrast, in the case of the egg case proteins, these motifs are interspersed throughout the sequence containing more than one motif in each repeat.

Varying properties were observed in the B10-22-17 recombinant of Library B, with an abundance of PySp1 from *L. hesperus* as well as the Flag from *N. clavipes*; the resulting protein had the highest percentage of β-turns as well as the highest change in β-sheet composition after methanol treatment (Figure 3A,C). The Flag protein found in prevalence in the B10-22-17 and B10-22-17_2_ constructs is theorized to have a high β-turn percentage due to repetitive GPGXX motifs [64]. PySp1 is theorized to be primarily constituted by random coils [39] which give piriform silk its flexibility. This would explain the large random coil presence in the B10 recombinant, in which the highest percentage of its molecular mass comprised PySp1 (Figure 3D). However, the addition of a PySp1_2_-AcSp1 unit to B10-22-17 decreased the β-turn percentage, while the β-sheet percentage increased (Figure 3A,C). This could be explained by the Flag protein now occupying a smaller percentage of the total protein mass, thus diluting the effects of the GPGXX motifs. This in conjunction with an increase in the β-sheet-favored *acsp1* gene could explain the reversal of secondary structures between these two recombinants. In both libraries, we can see that the amalgamation of different genes creates a family of varied and unique proteins, providing a source for a more diverse range of biomaterial designs.

Materials with a high Young’s modulus can withstand a large amount of stress before deformation, while those with a low Young’s modulus are elastic and deform easily. AFM analysis of three representative recombinant protein films, A10_2_, A261 and B10-22-17_2_, revealed that B10-22-17_2_ had a five-fold lower Young’s modulus when compared to A10_2_ and A261 (Figure 4B). A10_2_ protein films had a higher Young’s modulus including both the methanol induced and uninduced samples when compared to the corresponding A261 films (Figure 4B). A possible explanation for the data lies in the prevalence of elastic Flag motifs in B10-22-17_2_ which could contribute to a higher propensity for deformation compared to both A261 and A10_2_, which contained more rigid β-sheet prone motifs present in the *masp1*/*masp2* genes. It is also possible that the tendency of PySp1 to fold into random coil structures [40] and its high prevalence in B10-22-17_2_ could also play a role in the observed AFM results. Another observation that arose from the AFM results was the reduction in Young´s modulus after methanol treatment for A10_2_ and A261. A possible explanation is that the presence of blocks that did not lead to the formation of β-sheets intercalated between those that did, thus, disrupting proper β-sheet formation between modules. The topographical analysis from the AFM revealed that films generated using A10_2_ and B10-22-17_2_ showed uniform surfaces, while A261 films showed spherical structures on the surface; this may be due to the presence of air bubbles formed during the film deposition process or due to the presence of small aggregates of undissolved protein. 

Finally, in comparing the nanoscale AFM of the recombinant films to the macroscale Young’s modulus of the native spider silks fibers, both A10_2_ and A261 had higher Young’s moduli than all the native spider silk proteins. B10-22-17_2_ on, the other hand, had a lower Young’s modulus than native silk proteins except for Flag, which had a significantly higher Young’s modulus (Figure 5). However, as noted, the Young’s modulus measurements in the study are nanoscale measurements, whereas Young’s moduli measurements of native silks are based on macroscale Instron tensile testing [35]. Previous research comparing the mechanical properties of recombinant spider silk films to their corresponding fibers found that fibers were significantly stronger and had a higher Young’s modulus at the macroscale level compared to their corresponding films at the macroscale level [65]. Furthermore, when monomeric units of the repetitive core region were incorporated in the recombinant proteins, despite their relatively short sizes of 50–80 kDa compared to native silk, which can be up to 300 kDa in length, they nevertheless contained relatively large Young’s moduli at the nanoscale level. Whether these high Young’s moduli exist at a macroscale level is a topic for future study.

## 5. Conclusions

Functional spider silk-based biomaterials were produced from constructs generated through the random ligation of spidroin genes derived from the core repetitive region of *L. hesperus* and *N. clavipes*. Secondary structure characterization of the original recombinant constructs was carried out through FTIR. It was observed that on average, Library A recombinant proteins with AcSp1 and TuSp1 domains contained higher overall β-sheet compositions compared to Library B samples, possibly due to the presence of more poly A repetitive regions. Mechanical analysis using nano-indentation via AFM revealed that two representative Library A samples, A261 and A10_2_, both had a higher Young’s modulus compared to the representative Library B sample B10-22-17_2_. We speculate that the presence of β-sheet favored motifs in Library A and elastic β-helix and random coil motifs present in the Library B recombinant proteins contributed to the observed trends. The AFM results suggest that biomaterials with different mechanical properties were generated. This included Library A, resulting in robust materials with a high Young’s modulus and high stiffness and Library B containing biomaterials with a low Young’s modulus and low stiffness. Future studies should focus on the macroscopic qualities of fibers created from these recombinant proteins. With this work, we sought to generate new silk-like proteins that would broaden the mechanical properties of silks apart from those found in native silk proteins. From the observed trends of the recombinant proteins, the experiment led to the successful generation of proteins with varied mechanical properties expanding the diversity present in spider silks. This suggests that materials could be further utilized to expand and enhance the properties of spider silk proteins as future biomaterials.

## Figures and Tables

**Figure 1 materials-13-03596-f001:**
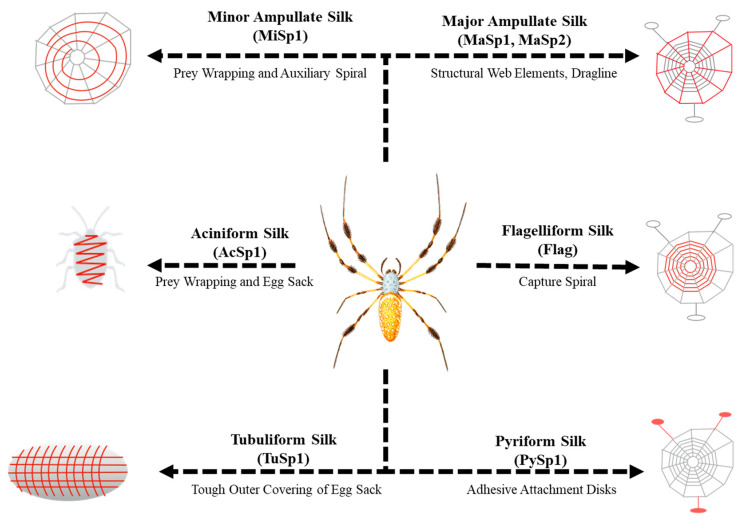
Summary figure of gene function and properties of spider silk. The function of each of the silks (shown in red) for the orb-weave specie *Nephila clavipes* are described.

**Figure 2 materials-13-03596-f002:**
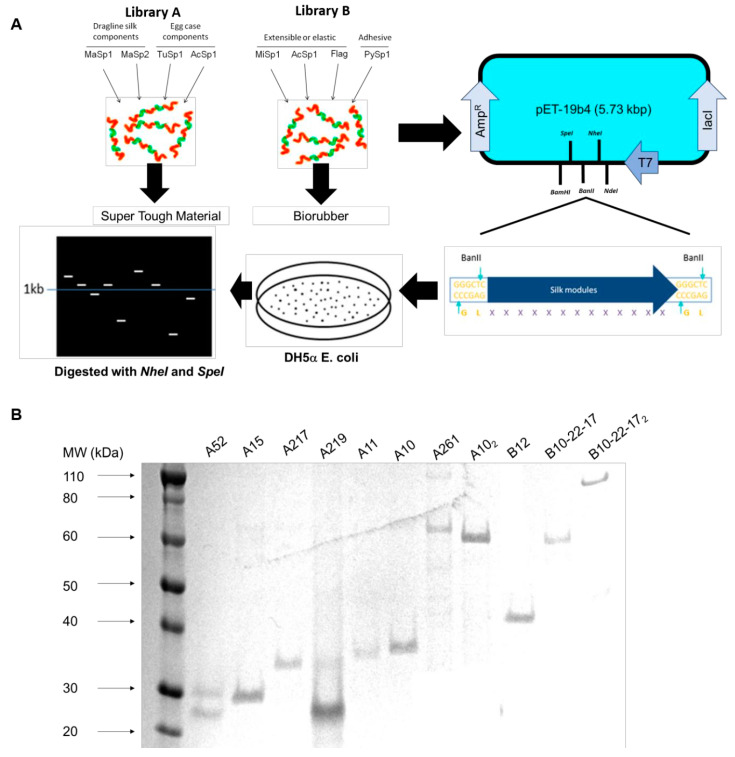
Establishment of the recombinant DNA libraries. (**A**) Generalized BanII restriction enzyme mediated cloning strategy for establishment of DNA library. (**B**) SDS-Page of select purified library constructs (A52, A15, A217, A219, A11, A10, A261, A102, B12, B10-22-17, B10-22-172).

**Figure 3 materials-13-03596-f003:**
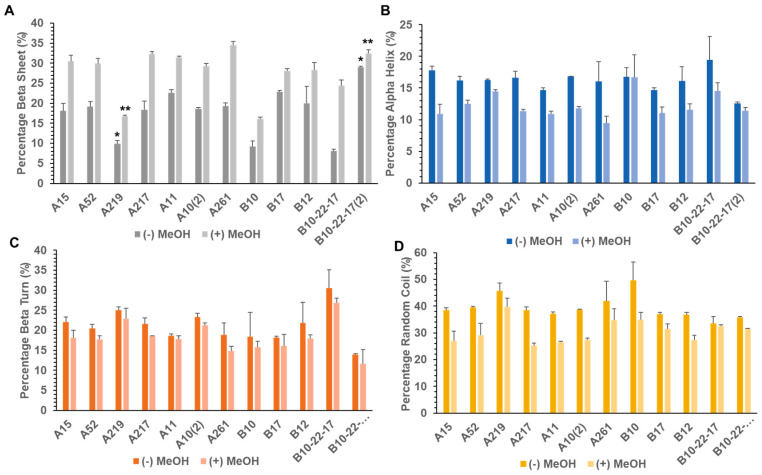
Secondary structure analysis of recombinant proteins in film format. Summary of percent of (**A**) β-sheet (dark gray before and light gray after methanol treatment), (**B**) α-helix (dark blue before and light blue after methanol treatment), (**C**) β-turns (dark orange before and light orange after methanol treatment) and (**D**) random coil (dark yellow before and light yellow after methanol treatment) of recombinant films. Percentage of the peak present at wavenumbers corresponding to a particular secondary structure across the amide I absorption band from 1720 to 1580 cm^−1^ (β-turns 1668–1696 cm^−1^, α-helix 1658–1667 cm^−1^, random coil 1630–1657 cm^−1^ and β-sheet 1618–1629 cm^−1^ and 1697–1703 cm^−1^) relative to the total peak area was used to determine the percentage secondary structure composition. Mean and standard deviation shown (n = 3 per sample). * and ** indicate statistical significance when compared to recombinants within the same library for both methanol treated and untreated samples. Significance was determined between recombinants within the same library via one-way ANOVA followed by a post hoc Tukey–Kramer test and within the same sample by a paired *t*-test (*p* > 0.05).

**Figure 4 materials-13-03596-f004:**
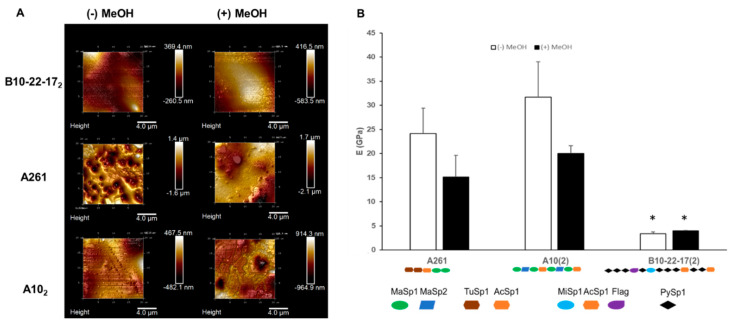
Mechanical characterization of selected protein films. (**A**) Representative Atomic Force Microscopy (AFM) topographic images of protein films with and without methanol exposure. (**B**) Summary graph of Young’s modulus (E) for the three recombinant proteins. Bars in black are methanol induced samples, while bars in white are non-methanol exposed samples (mean and standard deviation shown) (*) indicates significance from one-way ANOVA with a post-hoc Tukey test (*p* < 0.05).

**Figure 5 materials-13-03596-f005:**
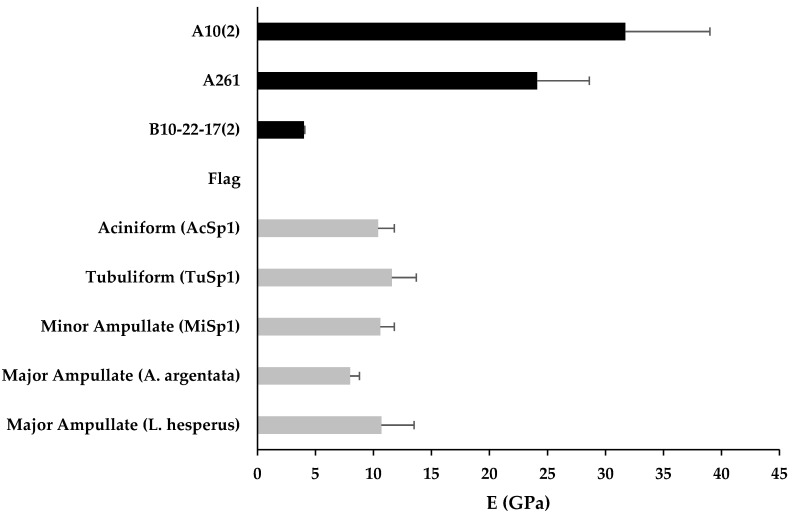
Summary graph showing Young’s modulus for uninduced Library A samples and methanol induced library B samples along with Young’s modulus for selected native proteins. Young’s moduli for native proteins derived from Instron tensile testing measurements of full-length fibers from [35]. Mean and standard deviation shown.

**Table 1 materials-13-03596-t001:** Summary table of primary sequences of individual monomeric protein constructs used in Library.

Module	Length (AAs)	Sequence (N’→C’)	Reference
MaSp1	35	GGAGQGGQGGYGQGGYGQGGAGQGGAGAAAAAAAA	[28]
MaSp2	40	GGSGPGGYGQGPAAYGPSGPSGQQGYGPGGSGAAAAAAAA	[28]
TuSp1	184	ASQAASQSASSSYSAASQSAFSQASSSALASSSSFSSAFSSASSASAVGQVGYQIGLNAAQTLGISNAPAFADAVSQAVRTVGVGASPFQYANAVSNAFGQLLGGQGILTQENAAGLASSVSSAISSAASSVAAQAASAAQSSAFAQSQAAAQAFSQAASRSASQSAAQAGSSSTSTTTTTSQA	[37]
MiSp1	42	GAGGYGQGQGAGAGAGAGAGAGGYGQGSGAGAAAGAAASAGA	[26]
AcSp1	188	FGLAIAQVLGTSGQVNDANVNQIGAKLATGILRGSSAVAPRLGIDLSGINVDSDIGSVTSLILSGSTLQMTIPAGGDDLSGGYPGGFPAGAQPSGGAPVDFGGPSAGGDVAAKLARSLASTLASSGVFRAAFNSRVSTPVAVQLTDALVQKIASNLGLDYATASKLRKASQAVSKVRMGSDTNAYALA	[62]
PySp1	40	ARAQAQAEAAARAQAQAEAAARAQAQAEAAARAQAQAEAA	[25]
Flag	40	GPGGAGPGGAGPGGAGPGGAGPGGAGPGGAGPGGAGPGGA	[41]

**Table 2 materials-13-03596-t002:** Summary table of relevant library colonies obtained from screening with molecular weights determined by primary sequence data and the corresponding gene sequence.

Library Colony	Estimated MW (kDa)	Sequence
A10	28.8	MaSp1-MaSp2-MaSp1-AcSp1
A11	29.3	MaSp2-MaSp1-MaSp2-AcSp1
A15	21.3	MaSp1-TuSp1
A46	21.9	MaSp2-TuSp1
A52	23.0	MaSp2-AcSp1
A217	27.4	(MaSp1)_3_-AcSp1
A219	22.0	(MaSp1)_4_-(MaSp2)_3_
A261	56.0	(TuSp1)_2_-AcSp1-(MaSp1)_2_
A10_2_	56.7	(MaSp1-MaSp2-MaSp1-AcSp1)_2_
B10	15.2	(PySp1)_3_-Flag
B12	32.7	PySp1-MiSp1-MaSp1-Flag-AcSp1
B17	27.6	(PySp1)_2_-AcSp1
B22	11.5	PySp1-MiSp1-PySp1
B10-22-B17	55.3	(PySp1)_3_-Flag-PySp1-MiSp1-PySp1_3_-AcSp1
B10-22-B17_2_	81.4	(PySp1)_3_-Flag-PySp1-MiSp1-PySp1_3_-((PySp1)_2_-AcSp1)_2_

**Table 3 materials-13-03596-t003:** Summary table of secondary structural motif counts determined from primary protein sequence for five select recombinant protein films (B10-22-172, A102, A261, A219, B10-22-17) along with their percentage secondary structure after methanol exposure determined from FTIR analysis.

Sample	Sequence	GPGXX Motifs	GGX Motifs (X = Y,L,Q)	A_n_/GA_n_ Motifs	AAQAA/AASQSA Motifs	Beta Sheet (%)	Beta Turn (%)	Alpha Helix (%)
B10-22-B17_2_	(PySp1)_3_-Flag-PySp1-MiSp1-PySp1_3_-((PySp1)_2_-AcSp1)_2_	8	4	0/4	0/0	32.5	11.6	15.5
A10_2_	(MaSp1-MaSp2-MaSp1-AcSp1)_2_	4	16	6/4	0/0	29.3	21.2	11.7
A261	(TuSp1)_2_-AcSp1-(MaSp1)_2_	0	9	2/2	2/4	34.4	14.9	9.5
A219	(MaSp1)_4_-(MaSp2)_3_	6	11	7/4	0/0	16.7	22.8	14.4
B10-22-B17	(PySp1)_3_-Flag-PySp1-MiSp1-PySp1_3_-AcSp1	8	3	0/4	0/0	24.3	26.9	14.5
Structural Role	 Elastic β-spiral	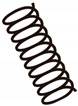 Amorphous 3_10_-helix	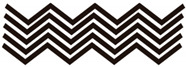 Crystalline β-sheet

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
