# Peer review of "Expanding Canonical Spider Silk Properties through a DNA Combinatorial Approach"

_materials, 2020, doi:10.3390/ma13163596_

Round 1

Reviewer 1 Report

I find the manuscript interesting and valuable for publication. In general, the work done in improving the properties of recombinant spider silk devices (films, fibers ecc...) is important and deserves to be investigated. 

Despite the overall good quality of the manuscript, I have major concerns in regard to the mechanical properties part.  Line 91-91: Thai phrase deserves better thinking. In general the properties of Flagelliform silk fibers are obtained by using entire catching spiral threads. However, it is also known that these are composed of both Aggregate and Flagelliform silk, where the former is essential in providing such extensibility (due to water collection and, thus, supercontraction). What are the author thoughts about this? I suggest having a read on the work done by Perez starting from Guinea et al., Biomacromolecules 2010, 11, 1174–1179.  Investigating the mechanical properties by means of nanoindentation with AFM is tricky and not trivial. One of the most important things to take into consideration is the fact that nanoindentation provides results that can be different with respect to the usual way of obtaining the mechanical properties. This is due to the asymmetrical nature of the material. Even though the authors mention that further experiments are going to be performed, I think that a control tensile test must be performed to check if the numbers are reliable (perhaps on strips cutted from the film, although I imagine the difficulties in obtaining the free standing film).  Moreover, there are several factors that should be taken into account and I do not see here any mention of these.   What is the tip radius of the AFM's probe?  What are the vacuum's values under which the samples have been dried?  The spacing between dots is crucial to avoid border's and small volume effects. In a square of 20x20 um^2 having a 16x16 grid could result in such a situation, especially when the penetration depth is not small in comparison to the spacing of the dots. Have the authors tried to test on a broader square or with less dots?   To indent a material that has a Young's modulus of the order of GPa, one wants to use a cantilever with a high stiffness. For example, in some tests that I have performed on silk we used a diamond coated cantilever with a stiffness of about 135 N/m. In this case the tip is way far too soft with respect to the common procedures, which is why more control tests (tensile ones) are crucial and necessary.  What are the values of the roughness on the analysed surfaces? This is another factor to take into account when performing nanoindentation.  The thickness of the film is very important. Normally the indentation depth must not reach the 10% of the thickness, otherwise one should consider also the contribution of the substrate. What is the thickness of the film in this case?  Normally the model used to compute the mechanical properties in nanoindentation is the Oliver and Pharr (J. Mater. Res., Vol. 7, No. 6, June 1992). It is thus important to partially justify the methodology. Is it possible to show some load displacement curves? If so, can the methodology be stated in a clearer way? Is it done by the softwarer of the AFM?  For polymers it is usual procedure to take v=0.2-0.3. Why did the author use 0.5?  Lines 285: "and nano-indentation studies were (missing words?) by AFM."
Lines 287-288: Are you sure that the relation between increased B-sheets and improved properties is true? Especially on behalf of what previously said.    For more input about nanoindentation I suggest to check this book (as an introduction): Fischer-Cripps, Nanoindentation, Springer, 2011   This is all. 

Author Response

Reviewer #1: I find the manuscript interesting and valuable for publication. In general, the work done in improving the properties of recombinant spider silk devices (films, fibers etc.) is important and deserves to be investigated.
Despite the overall good quality of the manuscript, I have major concerns in regard to the mechanical properties part. 

  1. Line 91-91: Thai phrase deserves better thinking. In general, the properties of Flagelliform silk fibers are obtained by using entire catching spiral threads. However, it is also known that these are composed of both Aggregate and Flagelliform silk, where the former is essential in providing such extensibility (due to water collection and, thus, supercontraction). What are the author thoughts about this? I suggest having a read on the work done by Perez starting from Guinea et al., Biomacromolecules 2010, 11, 1174–1179. 
  • Response: The authors agree with the reviewer that in native fibers there is a contribution of aggregate silk in the mechanical properties. However, in the work by Lewis in Adrianos et al., Biomacromolecules, 14(6), 1751–1760 using fibers created with synthetic Flag proteins, through mechanical testing analyses of the fibers it was demonstrated that the GGX motif present in the Flag protein contributes to the extensibility of the recombinant fibers. Thus, the results indicate that the motifs in flag are likely to supply mobility to the protein network of native clavipes flagelliform silk fibers.

  1. Investigating the mechanical properties by means of nanoindentation with AFM is tricky and not trivial. One of the most important things to take into consideration is the fact that nanoindentation provides results that can be different with respect to the usual way of obtaining the mechanical properties. This is due to the asymmetrical nature of the material. Even though the authors mention that further experiments are going to be performed, I think that a control tensile test must be performed to check if the numbers are reliable (perhaps on strips cutted from the film, although I imagine the difficulties in obtaining the free standing film). 
  • Response: We agree with the comment, however, relative differences among samples will still be relevant here as all materials were tested in a similar fashion. Also, given the low yield obtained in the purification process (50-100 mg/L), with we do not currently have sufficient material to conduct additional studies.  Due to the lab closure per Covid, this will not be possible for some time still. Most external facilities are also closed currently to outside users.

  1. Moreover, there are several factors that should be taken into account and I do not see here any mention of these.   What is the tip radius of the AFM's probe?  What are the vacuum's values under which the samples have been dried?  The spacing between dots is crucial to avoid border's and small volume effects. In a square of 20x20 um^2 having a 16x16 grid could result in such a situation, especially when the penetration depth is not small in comparison to the spacing of the dots. Have the authors tried to test on a broader square or with less dots?  
  • Response: Regarding tip radius, we use standard 5 μm diameter borosilicate glass beaded AFM probes and the radius is 2.5 μm. This information has been added to the text. The vacuum for drying the films was 25 mmHg. The measurements were performed over a 20 x 20 μm area using a 16 x 16 grid.

  1. To indent a material that has a Young's modulus of the order of GPa, one wants to use a cantilever with a high stiffness. For example, in some tests that I have performed on silk we used a diamond coated cantilever with a stiffness of about 135 N/m. In this case the tip is way far too soft with respect to the common procedures, which is why more control tests (tensile ones) are crucial and necessary.  What are the values of the roughness on the analysed surfaces? This is another factor to take into account when performing nanoindentation. 
  • Response: We do not have roughness values of the surfaces of the films. Please see the note above regarding the current inability to conduct additional experiments.

  1. The thickness of the film is very important. Normally the indentation depth must not reach the 10% of the thickness, otherwise one should consider also the contribution of the substrate. What is the thickness of the film in this case? 
  • Response: The thickness of the films was not calculated, however, similar past studies showed these films are 30 and 40 µm (e.g., Tucker et al. Biomacromolecules. 2014;15(8):3158-3170).

  1. Normally the model used to compute the mechanical properties in nanoindentation is the Oliver and Pharr (J. Mater. Res., Vol. 7, No. 6, June 1992). It is thus important to partially justify the methodology. Is it possible to show some load displacement curves? If so, can the methodology be stated in a clearer way? Is it done by the softwarer of the AFM? 
  • Response: The model used to calculate the mechanical properties, as stated in the methods section, is the Hertz model, that relies on the use of force-displacement curves. An example of load-displacement curve has been added to the supplemental material. A sentence has been added to the text: Nano-indentation was performed by collecting force-displacement measurements and a representative curve of the load-displacement curve is provided as Figure S3.

  1. For polymers it is usual procedure to take v=0.2-0.3. Why did the author use 0.5?
  • Response: Poisson’s ratio for the Young’s Modulus calculations using the Hertz model was 0.5 because protein networks like silk obey rubber elasticity. When analyzing soft samples, such as the majority of biological specimens, the nanomechanical behaviour is dominated by the elasticity. For soft rubber-like materials the Poisson ratio is generally set to 0.5 (Hawker et al., ACS Appl. Bio Mater. 2018, 1, 5, 1677–1686; JPK instruments application note, "Determining the elastic modulus of biological samples using atomic force microscopy", jpk.com). A sentence has been added to the methods section for clarification: "…because protein networks like silk obey rubber elasticity.".

  1. Lines 285: "and nano-indentation studies were (missing words?) by AFM."
  • Response: The word " carried out " has been added to the text.

  1. Lines 287-288: Are you sure that the relation between increased B-sheets and improved properties is true? Especially on behalf of what previously said.    For more input about nanoindentation I suggest to check this book (as an introduction): Fischer-Cripps, Nanoindentation, Springer, 2011   This is all.
  • Response: The main goal of the present work is to illustrate that we can expand and broaden the mechanical properties of silk-based biomaterials. In order to do that, we generated new protein variants by mixing different silk type domains. The goal was not to improve on silk material properties but to generate an amalgam of new variants with novel properties that represent hybrids of those properties already known.

Reviewer 2 Report

This reviewer's comments are mentioned in the attached file.

Author Response

Reviewer #2: In the present study, the authors synthesized a wide variety of spider silk-like proteins through a DNA combinatorial approach using E.coli expression system. This reviewer thinks that the concept and content of the present work should be appropriate for this Journal Materials. However, this study lacks some experimental data and thus needs to be published after major revisions.

The reviewerʼs comments are mentioned as follows:
1. There is no description about the yield of the expressed protein. Because the authors mentioned that the main objective of the present study is to synthesize the proteins with varied mechanical properties to use as biomaterials, it is important to mention the yield of the resultant protein.

  • Response: The yield of the proteins was 50 to 100 mg/L. A sentence has been added to the text, line 241-242: The yield of the expressed proteins ranged between 50-100 mg/L.

  1. The authors did not present raw data of the FT-IR spectra. In addition, the deconvolution graphs of the FT-IR data, which were used to estimate the percentage of the secondary structure, will help the reader for better understanding of the secondary structure analysis shown in Figure 3. This reviewer does not understand how the authors analyzed the FT-IR data and evaluate the secondary structure.
  • Response: A representative FT-IR spectra for recombinant A102 protein across amide I and amide II absorption spectra of methanol treated and untreated films has been added as supplemental Figure 1. To help better understand the deconvolution procedure on the secondary structure analysis, representative images showing the selected gaussian peaks for recombinant A102 across amide I and amide II absorption spectra of untreated and methanol treated films has been added as Supplemental material Figure 2. A sentence has been added to the text: A representative FTIR spectra for untreated and treated A102 films is provided in Figure S1, as well as a representative deconvoluted spectrum for untreated and treated A102 films in Figure S2.

The relative area of the curves corresponding to each of the secondary structural motifs was calculated; relative areas were divided by the summed area of all the deconvoluted curves and transformed into percentages. A sentence has been added to the FT-IR methods section: Using the second derivative method, the relative area of the curves corresponding to each of the secondary structural motifs was calculated; relative areas were divided by the summed area of all the deconvoluted curves and transformed into percentages.

  1. The authors should present raw data (load ‒ displacement plot) of AFM nano-indentation study. In addition, this reviewer strongly recommends conducting a tensile test analysis to compare the mechanical properties of the obtained film with the mechanical parameters obtained in the previous studies.
  • Response: For the load-displacement curve, an example of the curves generated has been added to the supplemental material; a sentence has been added to the text: Nano-indentation was performed by collecting force-displacement measurements and a representative load-displacement curve is provided as Figure S3.

Regarding the tensile test, we do not currently have sufficient material to conduct additional studies.  Due to the lab closure per Covid, this will not be possible for some time still. Most external facilities are also closed currently to outside users.

  1. (page 9 line 290) Why did the surface of silk films show spherical structure before methanol treatment in the case of A261? The authors should clear this in the Discussion section.
  • Response: The fact that we observed spherical structures on the surface of silk films can be due to the presence or air bubbles formed during the deposition process or due to the presence of small aggregates of undissolved protein. A sentence has been added to the discussion section to address this point: The topographical analysis from the AFM revealed that while films generated using A102 and B10-22-172 showed a uniform surface, A261 films showed spherical structures on the surface of silk films; possibly due to the presence or air bubbles formed during the deposition process or due to the presence of small aggregates of undissolved protein.

  1. (page 9 line 294) General understanding in the field of silk is that the methanol treatment induces the b-sheet structure and increase the Youngʼs modulus of the material. However, the Youngʼs modulus showed decrease after methanol treatment although the b-sheet content increase. The authors should explain this in the Discussion section.
  • Response: Since the proteins we are generating are constituted by different silk types and not uniform monomers, and even though the β-sheet content increases after methanol treatment, the presence of blocks or domains that does not exclusively lead to the formation of β-sheets may be possible. This may be due to entrapped or intercalated domains between those responsible for β-sheet formation, thus a reduction in Young´s modulus. A possible explanation is that these domains may be disrupting proper β-sheet interactions between modules, acting as spacers. A sentence has been added to the discussion section: Another observation that arose from the AFM results was the reduction in Young´s modulus after methanol treatment for A102 and A261. A possible explanation is that the presence of blocks that do not exclusively lead to the formation of β-sheets, possibly intercalated between those responsible for that secondary structure, thus disrupting proper β-sheet interactions between modules.

  1. (Figure 5) Although the authors assigned (‒) MeOH group in the left column in Figure 3, the authors assigned (‒) MeOH group in the right column in Figure 5. This can cause misunderstanding of the readers. The authors should be very careful about the consistency of the data presentation.
  • Response: The figure has been modified to be consistent with previous figures.

  1. There are so many miswriting in the text, which downgrades the quality of the manuscript. This
    reviewer strongly requests the authors to doublecheck the manuscript and use English editing service. The following points should be addressed:
    A) (page 1 line 26) Delete the extra space.
    B) (page 1 line 28) Delete the extra “of”.
    C) (page 1 line 33) Delete the extra space.
    D) (page 1 line 42) Put the period instead of comma.
    E) (caption of Figure 2) a) and b) should be capital letters.
    F) (page 8 line 247) “1720-8.1) 1580 cm-1” does not make sense.
    G) (page 9 line 285) Put “conducted” after “were”.
    H) (page 10 line 327) Delete “from”.
    I) (page 10 line 335) Delete extra space.
    J) (page 11 line 350) Put “of” after “range”.
  • Response: All the points have been addressed in the text and have been highlighted in yellow.

Round 2

Reviewer 1 Report

In general, I am satisfied with some answers, but the mechanical part seems not to be solid.

To reply point 3-4)

Are the authors sure that the RADIUS (and not the height) of the tip’s probe is 2.5 um? If the answer is yes, I would like to know the brand, since these radii are generally between 5 and 50 nm.

Moreover, the authors state that they use a grid of about 20x20 cm with 16x16 points. This means that each point is 1.25 um distant from another, which gives not reliable data of mechanical properties due to the overlapping of the holes (since the tip is 2.5 um, and the depth is 0.750 um). In this context, the data obtained are not to be considered solid.

Moreover, before doing any indentation experiments it is common procedure to perform topological imaging to calculate the roughness. This affect a lot the numbers, and without it (that can be easily of the order of the um, even though in this case is unlikely) the analysis is even weaker. Furthermore, when performing AFM nanoindentation experiments, you should first take a topographical image of the surface. So, it would be just a matter of data analysis (for example with gwyddion, open source software) to get the roughness. If the authors do not have any image of the surface, it means that they landed randomly the tip, which is very unlikely.  

To reply point 5)

The answer is not satisfactory. You cannot claim that the thickness if your film is a number if you refer to a protocol not even performed in the same experimental group (and not even with the same protocol). You can claim that your film is similar, but at least a rough investigation is needed.

To reply point 6)

The authors have not replied to the question: is the analysis performed manually (Excel or Origin) or by means of a software (Nanoscope analysis for example)?

Is the displayed curve a backward or forward curve (retraction or approach of the probe)? They both should be displayed.

To reply point 9)

To state that you broad the mechanical properties of silk-based biomaterials one should be first sure about the method used to obtain such properties.

To conclude, the paper is interesting, but the mechanical part reveals a deep lack in methodology and awareness of the common nanoindentation procedures. For this reason I recommend major revision, or even directly remove such part (which is not self-standing).  

Reviewer 2 Report

The manuscript “Expanding canonical spider silk properties through a DNA combinatorial approach”, submitted by Jaleel et al. was revised according to the reviewer’s suggestions. The authors addressed points of concern and included the respective changes resulting in improved data interpretation and increased comprehensibility. Although considering the single aspects of the paper might lead to the impression, that the findings presented here are only partially novel, the extend of the study and its holistic approach are good and will give a valuable contribution to the scientific field of material science. I therefore recommend this manuscript for publication.

Author Response

none required